# Design of Electromagnetic Control of the Needle Gripping Mechanism

**Jiří Komárek *** and **Vojtěch Klogner**

Department of Textile Machine Design, Faculty of Mechanical Engineering, Technical University of Liberec, 461 17 Liberec, Czech Republic; vojtech.klogner12@gmail.com
* Correspondence: jiri.komarek@tul.cz; Tel.: +420-485-353-418

**Abstract:** This paper deals with the modification of the mechanical system of the needle bar. The purpose of this work is to reduce the vibration and noise of the sewing machine for creating a decorative stitch. A special floating needle is used to sew this stitch, in which two mechanical systems of needle bars handover through the sewn material, so that a perfect imitation of a hand stitch is created. The original system, which controls the release of the needle at the handover location by abruptly stopping the needle bar control element, could be replaced by a new system that uses magnetic force to release the needle. In addition to the usual design procedure, numerical simulations of the attractive force of the electromagnet are also used in the design of a suitable electromagnet. At the same time, an electrical circuit is also designed to allow the needle to be released and gripped quickly. The advantages of the new system lie not only in reducing vibrations and the associated increase in the operation speed of the machine, but also in making it easier for the machine to switch to possible automated or semi-automated production.

**Keywords:** sewing machine; needle bar; floating needle; electromagnet; electromagnetic simulation; noise reduction





## 1. Introduction

The subject of the research is a special sewing machine, which is used to sew a decorative stitch. To do this, it uses a so-called floating needle, which has a tip on both sides and an eye in the middle. This needle passes through the sewn material back and forth with each stitch, thus imitating the hand stitch very well. In this process, two mechanical systems of needle bar (hereinafter also referred to as the needle bar), located one above and the other below the machine's worktop, transmit the needle. The release of the needle during its transfer is controlled on the needle bar by a mechanical stop. Excessive vibration and noise occur when the needle bar control element hits the stop [1]. Modifications to the existing mechanism to reduce noise have been discussed before and the results have been published in [2,3].

The principle of imitation of a hand stitch, which is used by the investigated sewing machine, has been known for a long time. The first mention of a needle with spikes at both ends appeared in 1755. At the time, it was a proposal by the German inventor Charles F. Weisenthal, who wanted to replace low-productivity hand sewing with machine sewing [4,5]. Long after the discovery of the sewing machine with a chain stitch, and later also with a lock stitch [6], Weisenthal's needle again found application in hand-stitch imitation machines. The principle of making a hand stitch was invented by Jessie Langsdorf in 1936. His patent, issued in 1937, was used by the Naftali brothers in collaboration with the AMF (American Machine and Foundry Company, Brooklyn, New York, NY, USA) in the manufacture of tie-making machines [7,8]. In the following years, the system was gradually improved and found its use in decorative sewing machines and button sewing machines. Since then, the sewing machine has undergone several modifications, but this system is still unsatisfactory.

Modern sewing machines must meet many requirements, such as quiet operation, minimal vibration [9], long life of the mechanisms used and easy operation of the machine. Great emphasis is also placed on reducing the time of the sewing process, which has a direct effect on increasing sewing productivity [10]. New work is also based on these assumptions, which deals with the design of a completely new system that does not require a mechanical bond to release the needle. This will allow the noise generated by the release of the needle on the original system to be completely eliminated. The motivation for this research is, on the one hand, the great pressure of the consumer industry to reduce production costs and, on the other hand, the need to protect the working operator of the sewing machine from higher noise levels [11–13].

The mechanical system of the needle bar performs a rectilinear reciprocating movement, which is realized on the original machine by a cam mechanism. According to previous studies, it has been found that replacing an existing needle bar drive with a controlled servo drive will also significantly affect vibration reduction, including meeting the assumption that the operating speed of the machine will increase. The design of a new needle attachment system should also contribute to this. In addition, replacing the original needle bar drive with a mechatronic system should facilitate the transition to automated production [14–17]. Thus, there is a noticeable support for automation in the field of the clothing industry, where there is an effort to at least partially automate the process of manufacturing clothing. It is quite certain that if the sewing process is to remain reliable after increasing the operating speed, the mechanical system of the needle bar needs to be adjusted.

## 2. The Current Method of Gripping the Needle

The section of the mechanical system of the needle bar is shown in Figure 1. The needle bar performs a rectilinear reciprocating motion. This is ensured by the carrier 12, which is connected to the needle bar shell 4 and drives the needle bar according to the required stroke, which is generated by the cam mechanism on the original sewing machine.

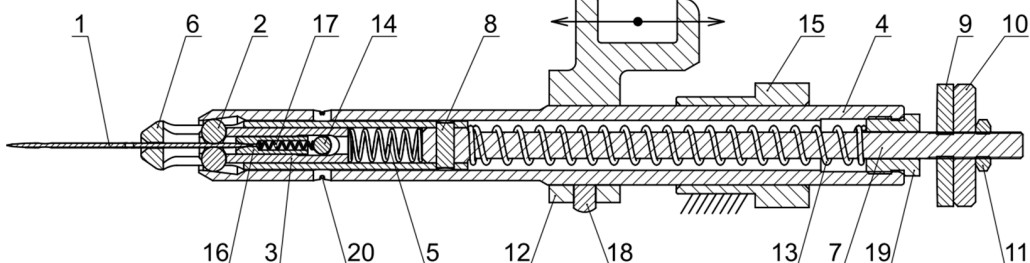

**Figure 1.** The current method of gripping the needle—gripping the needle.

The floating needle 1 is held in the needle bar by gripping system 2. The system consists of a pair of balls which are housed in a roller 3. Figure 1 shows the mechanical system of the needle bar holding the needle, where the balls are pressed against the conical surface by means of springs 5, and 13, which causes them to be pinched.

The release of the needle is controlled by the impact of the control element (which consists of parts 6, 7, 8, 9, 10, and 11) on the stopper 15, located on the machine frame. The release occurs during the needle bar movement to the needle handover location. This condition is shown in Figure 2.

The original mechanical system of the needle bar is designed for lower speed, and therefore at higher speed the inner parts of the needle bar oscillate, which is described in detail in [18,19].

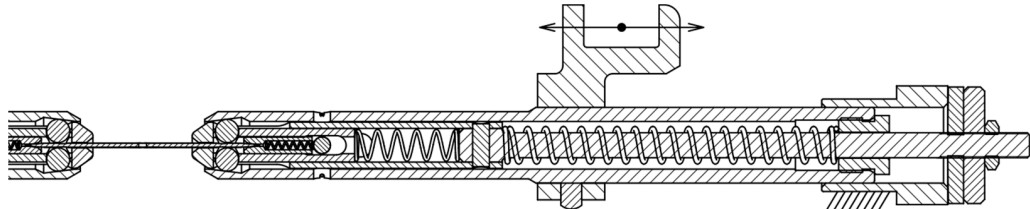

**Figure 2.** The current method of gripping the needle—releasing the needle.

## 3. Design of a New System

### 3.1. Requirements for a New Mechanism

During the sewing process, forces act on the needle which affect the gripping part of the needle bar. A measuring aid was designed to determine these forces. It was a modified stitch plate, which was equipped with strain gauges. The measurement of the forces acting on the needle during sewing on the investigated sewing machine is described in detail in [20]. This measurement determined the forces with which the needle should be held during the sewing process. The needle must be held with a minimum force of 25 N when piercing. The force must be at least 15 N during drawing through the sewn material. The needle must be flexible during transfer. The weight of the needle bar should not increase significantly so as not to negatively affect the required torque of the drive motor.

One of the solutions to simplify the mechanical system and at the same time eliminate the problematic impact of the control element into the stop is to use the magnetic force to attach the needle using an electromagnet [21]. The general advantage of electromagnetic control is that it allows the power supply parameters to be changed independently, regardless of the other mechanisms of the sewing machine, because they are not connected by any mechanical coupling. Another advantage is that the electromagnets can be overloaded for a short time, which allows their more efficient use. A relatively small stroke, approximately 2 mm, is required to operate the existing needle gripping system. For such small strokes, the use of electromagnets is a suitable choice due to their attractive force characteristics [22].

There are a number of arrangements, or constructions, of the electromagnet. The most basic electromagnetic arrangement is the front electromagnet, which is shown in Figure 3. It excels in high strength in the attracted state. However, as the air gap increases, its strength decreases very sharply.

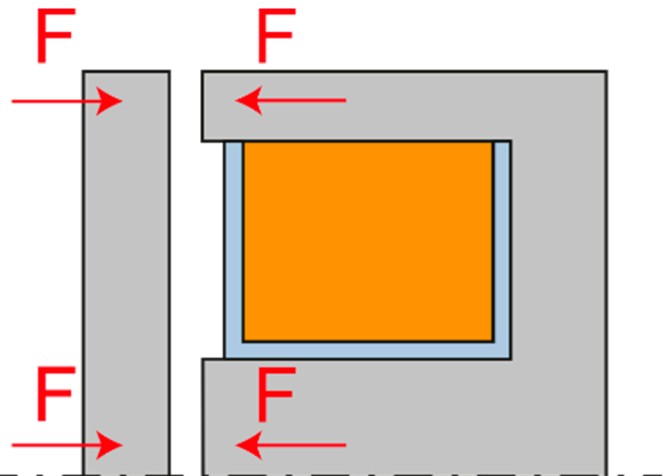

**Figure 3.** Front electromagnet.

Another variant is the so-called solenoid, where the armature is pulled by an electromagnet. This solution is characterized by a more gradual decrease of the attractive force. It is therefore suitable for applications in which it is necessary to apply force at different anchor positions. The electromagnetic solenoid can exist in two variants, pushing and

pulling, according to the direction of the derived arm force. The pushing arrangement shown in Figure 4 has a flatter force characteristic, i.e., in the position of the zero air gap, it has a smaller attractive force, and on the contrary, in the position of the maximum air gap, it has a greater attractive force. The pulling arrangement is shown in Figure 5.

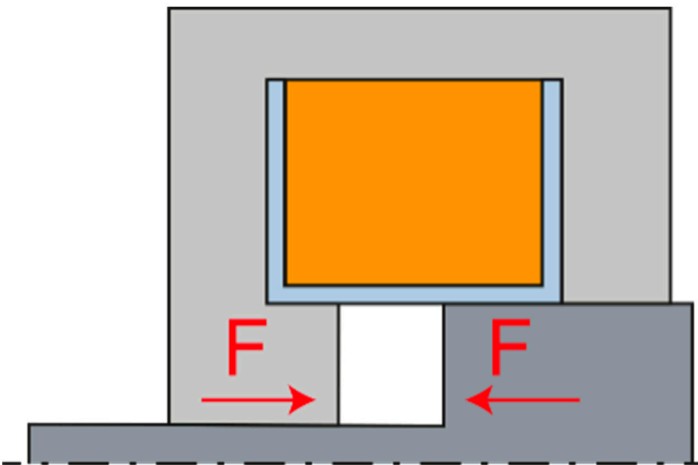

**Figure 4.** Push solenoid.

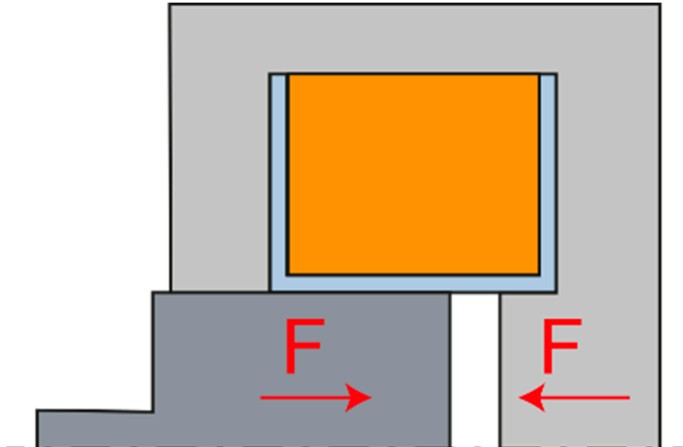

**Figure 5.** Pull solenoid.

In general, the magnetic force depends on the perpendicular cross-sections of the surfaces that make up the magnetic circuit, i.e., the larger diameter armature has a greater attractive force than the smaller diameter armature while maintaining the coil parameters.

Another possibility is to use the needle itself as the armature of the electromagnet and, unlike the previous two variants, to omit the ball gripping mechanism, as shown in Figure 6. However, due to the very small needle diameter, approximately 1 mm, the attractive force would be very small. This would not be a problem when piercing, but without a self-holding ball gripping mechanism, the needle could be easily pulled out of the gripper. This variant would therefore only be suitable for sewing delicate fabrics where there are no large frictional forces during drawing.

Two solenoid layout options allow us to design two variants of the mechanism. In the first, when the electromagnet is not energized, the spring closes the mechanism and keeps the needle gripped. In the second, when the electromagnet is not energized, the spring opens the mechanism. From the point of view of the functionality of the mechanism, it would be more appropriate to use the variant where the needle remains gripped and does not fall out spontaneously when the power supply is disconnected. However, it is

necessary to perform a force analysis to select a more suitable variant for the dynamic characteristics of the needle bar.

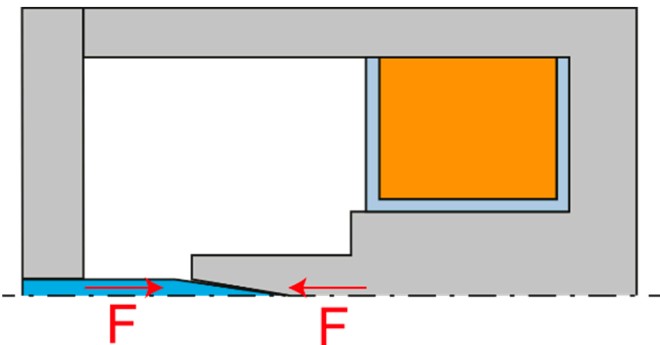

**Figure 6.** The needle itself as the armature of the electromagnet.

For an initial consideration of electromagnet design, we can specify which forces act on the control element. Inasmuch as we require the control element to move at the same speed and acceleration as the needle bar shell most of the time, we can define the required force that will act on the control element. Because the acceleration of the needle bar is precisely defined by the stroke dependence, we can determine the course of this inertial force relatively accurately.

Another force present is the force exerted by the spring, which returns the mechanism to its initial position. We assume that the spring has a linear stiffness characteristic. In the graph in Figure 7, it should have a general force function in the transition area, but for simplicity it is shown as linear with respect to the rotation of the control cam [23]. The last significant force is the attractive force of the solenoid. The force characteristic of the solenoid is hyperbolic depending on the actual size of the working air gap.

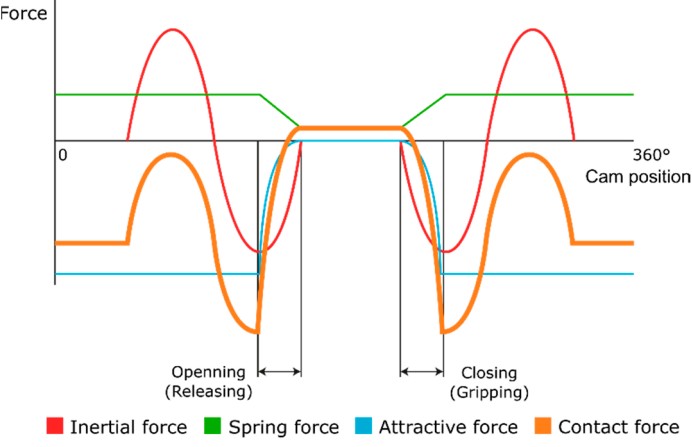

**Figure 7.** The course of forces on the mechanism with the push solenoid.

We can also show the contact force between the control element and the needle bar shell. If we want the control element to be in a clearly defined position, the contact force must be negative for the closed state and positive for the open state. Contact force can be zero with no contact. During needle release, this force determines the acceleration of the control element relative to the needle bar shell. This phenomenon is very important as it determines the time for which the needle will be released or attached. When analysing the force characteristic, the time at which the gripping mechanism is closed or opened is important. For the purpose of simplification, in this case, this time is chosen in the sinusoidal acceleration amplitude. If we plot all the forces in the graph in Figure 8 as a function of the cam rotation, we can observe several important findings.

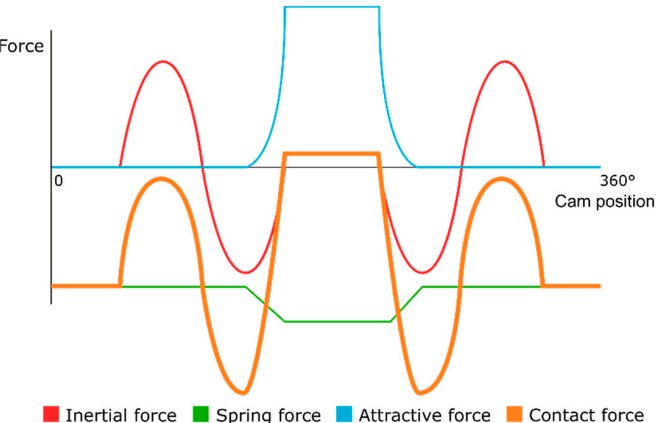

**Figure 8.** The course of forces on the mechanism with the pull solenoid.

In the case of a solenoid in the pushing arrangement, its applied force in the attracted state must be greater than the sum of the force exerted by the spring and the amplitude of the inertial force so as not to disengage the control element from the needle bar shell. In other words, the total force acting on the control element must be less than zero in the interval when the required needle gripping and greater than zero in the interval when the needle is released.

If we compare the variant of the push and pull solenoid, we find that there are a number of differences. The fundamental difference is that the pull solenoid only exerts force when the mechanism is in the open state. This means that in order to compensate for the accelerating force when moving the needle bar, there is only a spring, which must therefore exert several times more force than in the case of the push solenoid. Assuming that the working stroke is the same for both springs, this spring would certainly have larger installation dimensions and greater weight. The second problem is that the pull solenoid has the least force at the largest air gap, that is, at the exact moment when we need to open the gripper quickly.

For the following reasons, the variant with the push solenoid was chosen for the solution. The acceleration of the needle bar only occurs when the needle is gripped, i.e., at the moment when the push solenoid exerts the greatest force. In addition, due to the shape of the armature, the push solenoid exerts more force than the pull solenoid at the moment of the largest air gap. Nevertheless, there is one disadvantage. When the power supply is disconnected, the needle is released by the return spring.

Figure 9 schematically shows the new needle bar design. The needle bar shell 1 must be made of several pieces in order to be able to assemble the mechanism. The needle 2 is held by balls 3 housed in a roller 4. The roller is controlled by a part 9, which is screwed onto the armature of the electromagnet 11. The armature is returned to its initial position by a spring 8. The electromagnet itself consists of a coil 13 and a frame 12. The movement of the armature is damped by a rubber pad 10. To seat the needle, there is a roller 5, which is returned by the spring 6 to the basic position. The spring is supported on a pin 7, which is fixed in the needle bar shell.

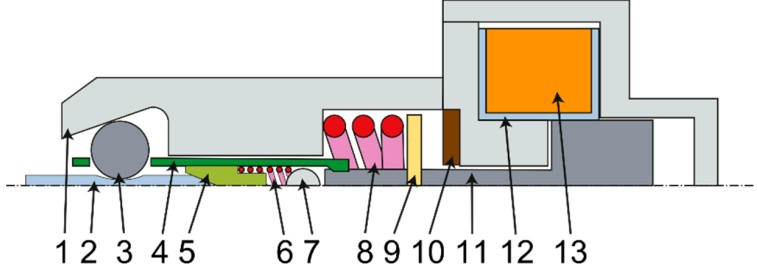

**Figure 9.** Design of the new mechanical system of the needle bar.

### 3.2. Winding Dimensioning

The design of the electromagnet winding is a relatively complex issue that solves the magnetic fluxes around the coil, the heat dissipation from the coil space to the surroundings also plays a big role here. The usual procedure in the design of electromagnets is the use of empirical relationships, which are based on simplifying assumptions, especially discretization of a continuous problem into several material blocks. Here, the methodology of Professor Cigánek is used [24].

The calculation is based on the empirical equation of the attractive force *F* of the electromagnet:

$$F = \pi 10^5 B^2 d^2 \varepsilon^2 (1 + \nu) \tag{1}$$

including the induction in the centre of the coil *B*, the armature diameter *d*, the magnetic flux deflection factor $\varepsilon$ and the flux increase factor $\nu$.

For an approximate calculation of the armature diameter, we can use this equation:

$$d = 0.115 \sqrt[5]{\frac{F}{1 + \nu} \delta^2} \tag{2}$$

in which only the required attractive force *F* occurs at the required air gap $\delta$.

For a more accurate calculation, it is then necessary to use this equation:

$$d = \sqrt[5]{\frac{20.2 \times 10^5 \rho k_s^2 (1 + \beta) F \delta^2}{\xi \beta \varepsilon^2 \lambda (1 + 2\beta)(1 + \nu) \alpha \Delta \vartheta_m}} \tag{3}$$

in which there are other auxiliary factors and selected parameters. Specifically, the ratio of winding thickness to armature diameter $\beta$, length ratio $\lambda$ to armature diameter, coil winding factor $\xi$ (i.e., how much geometric space of the coil is occupied by the copper itself), conductor resistivity at operating temperature $\rho$, magnetic circuit saturation factor $k_s$, heat conduction coefficient $\alpha$, and finally the temperature difference between the operating temperature and the environment $\vartheta_m$.

Figure 10 shows the course of the coil weight (green-yellow area) and the armature weight (purple area) depending on the parameters $\beta$ and $\lambda$. It is possible to see the opposite trends of both dependencies, so it is necessary to select values that will compromise the requirements for the weight of the armature and coil.

The calculation of the factors $\nu$ and $\varepsilon$ can be performed using the following relations, although these relations contain an average *d*, which is currently unknown. It is either possible to use an approximate relation to calculate the mean *d*, or to calculate these three equations iteratively until the deviation is acceptably small. In practice, the average *d* is chosen as an integer for production reasons, so accuracy does not matter much at this stage.

$$\nu = \frac{2\delta^2}{d^3 \varepsilon^2 \ln(1 + 2\beta)} \tag{4}$$

$$\varepsilon = 1 + \frac{\delta}{d} - \left(\frac{\delta}{d}\right)^2 \tag{5}$$

This relation applies to the cross section of conductor *S*:

$$S = \frac{4480}{U\varepsilon} k_s (1 + 0.75) \rho \delta \sqrt{\frac{F}{1 + \nu}} \tag{6}$$

The number of turns of coil *N* is calculated as follows:

$$N = \frac{\xi}{S} \beta \lambda d^2 \tag{7}$$

The approximate value of the induction in the centre of the coil $B$ can be expressed by the relation:

$$B = \sqrt{\frac{F}{\pi 10^5 d^2 \varepsilon^2 (1 + \nu)}} \tag{8}$$

The last equation needed for the calculation is the steady state equation of the flowing current $I$:

$$I = \frac{10^7 B \delta k_s}{4\pi N} \tag{9}$$

The result of these empirical equations are the calculated values of armature diameter $d$ (respectively radius $r1$), coil length $L$ and coil diameter $D$ (respectively radius $R$), coil conductor diameter $d_C$ and current $I$ flowing through the coil at a given voltage. Other values that need to be selected are related to the design of the electromagnet itself, especially the air gap between the components.

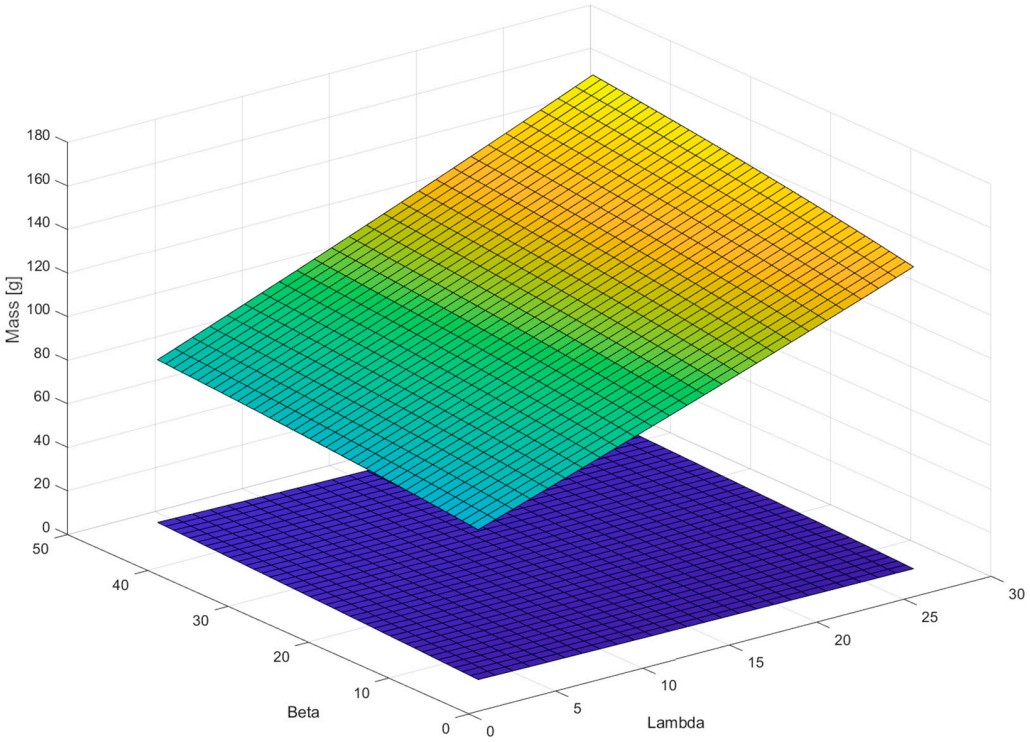

**Figure 10.** Graphical representation of the dependence of the coil and armature weights on the dimensional ratios.

The calculation performed should be taken as a guide only, as it applies to the pull solenoid, not the push solenoid, which has a different design and different magnetic conditions. Nevertheless, it provides enough input information to create an electromagnetic simulation that calculates the attractive force much more accurately.

Table 1 shows the reference parameters. Table 2 shows the main calculation parameters of the coil.

In Figure 11, it is possible to see the proposed dimensions that are needed to create a simulation model of the attractive force; the values of the dimensions are entered in Table 3. The table contains the construction dimensions of the electromagnet, which are based on the technological possibilities of electromagnet production and the construction arrangement of the needle bar. The basic parameters obtained on the basis of the calculation given in Section 3.2 were respected during the design.

**Table 1.** Reference parameters.

| Parameter | Symbol | Value |
|---|---|---|
| Attractive force at maximum stroke | $F$ | 6 N |
| Air gab at maximum stroke | $\delta$ | 2 mm |
| Coil voltage | $U$ | 3 V |
| Coil winding factor | $\xi$ | 0.7 |
| Magnetic circuit saturation factor | $k_s$ | 1.3 |
| Flux increase factor | $\nu$ | $1.41 \times 10^{-5}$ |
| Magnetic flux deflection factor | $\varepsilon$ | 1 |
| Length ratio to armature diameter | $\lambda$ | 2 |
| Ratio of winding thickness to armature diameter | $\beta$ | 0.4 |

**Table 2.** Calculated parameters.

| Parameter | Symbol | Value |
|---|---|---|
| Armature diameter | $d$ | 12 mm |
| Coil diameter | $D$ | 25.6 mm |
| Coil length | $L$ | 28.8 mm |
| Coil conductor diameter | $d_C$ | 0.6 mm |
| Magnetic induction in the centre of the coil | $B$ | 0.36 T |
| Number of turns of coil | $N$ | 400 |
| Static current | $I$ | 1.9 A |

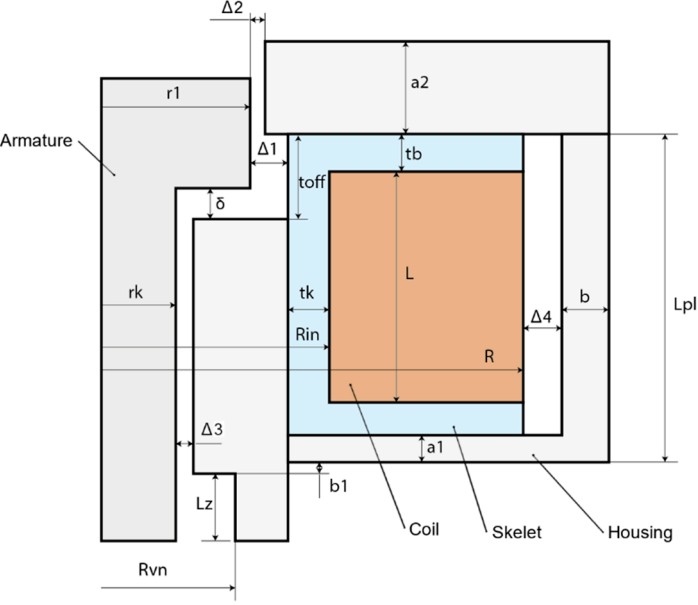

**Figure 11.** Electromagnet dimensions.

**Table 3.** Electromagnet dimensions.

| Parameter | Symbol | Value |
|---|---|---|
| Air gap | $\delta$ | 0–2 mm |
| Internal air gap 1 | Δ1 | 1 mm |
| Internal air gap 2 | Δ2 | 0.1 mm |
| Internal air gap 3 | Δ3 | 0.1 mm |
| External air gap | Δ4 | 0.1 mm |
| Shaft radius | rk | 2 mm |
| Armature radius | r1 | 6 mm |
| Coil inside radius | Rin | 8 mm |
| Coil outside radius | R | 12.8 mm |

**Table 3.** *Cont.*

| Parameter | Symbol | Value |
| --- | --- | --- |
| Skeletal thickness radial | tk | 1 mm |
| Skeletal thickness axial | tb | 1 mm |
| Coil length | L | 28.8 mm |
| Housing thickness 1 | a1 | 7 mm |

*3.3. Simulation of the Attractive Force of an Electromagnet*

As already mentioned, the empirical design serves only to approximate the dimensions and is, today, no longer sufficient due to its low accuracy. At present, it is possible to use numerical electromagnetic simulation based on the finite element method. There are several pieces of software that deal with this issue; the program FEMM 4.2 [25,26] was chosen for this work, because it has an integrated interface to the Matlab environment, which continues to work with data.

The computational model is static, i.e., the entire magnetic circuit is in a steady state. This is advantageous for simulating the end positions in which the electromagnet remains. In this case, however, we are also interested in the state between the end positions, which can be achieved by a quasi-static problem, i.e., by dividing into discrete time steps in which the circuit is static. This introduces a certain inaccuracy into the model, but this procedure is sufficient to determine the attractive force characteristic.

As a result of the electromagnetic simulation, we are particularly interested in the characteristics of the attractive force of the electromagnet, i.e., its dependence on the size of the air gap. The second important datum is the total inductance of the electromagnet, which is important for the design of the electrical circuit of the coil supply.

In the simulation, we discretize the whole problem into a so-called finite element network; the software can create this network itself based on the defined geometry and parameters, such as the maximum element size. The network element type is set to a triangle, the minimum angle of the triangle is set to $30°$, and the required accuracy is set to $1 \times 10^{-8}$.

The boundary condition is set to a free environment whose radius is 120 mm, as can be seen in Figure 12.

Furthermore, it is necessary to define the materials of individual bodies because during magnetization, it is necessary to know the relative permeability and hysteresis B-H curve. The coil is defined by the resistivity of the conductor, the cross section, the number of turns and the current flowing.

The material of the core, hence the metal casing of the coil, must be chosen with regard to its magnetic properties. In general, relative permeability affects the magnetic flux, so the greater the relative permeability, the less resistance of the material to the passing magnetic flux. The shape of the hysteresis B-H curve affects the magnetic losses or the inductance of the entire electromagnet.

Magnetically soft steels are used as core materials, i.e., steels which, after the elimination of the magnetic field, do not remain magnetized, so that their attractive force is close to zero. These steels consist of either almost pure iron or, conversely, a very high alloy content. For comparison, two steels were selected that have all of their parameters defined in the FEMM 4.20 program, namely AISI 1010, which has an iron content of 99.5% and a steel called Supermalloy, which contains 5% iron, 79% nickel and 5% molybdenum.

The FEMM simulation program can be controlled using commands in the Matlab script, which allows us to iterate between the individual variants very quickly, as everything can be automated. The quasi-static state is achieved by gradually shifting the geometry of the anchor, i.e., by reducing the air gap. At each state, the attractive force is calculated, which is equal to the volume integral of the Lorenz force.

$$F_{lorenz} = \int J \times B dV \qquad (10)$$

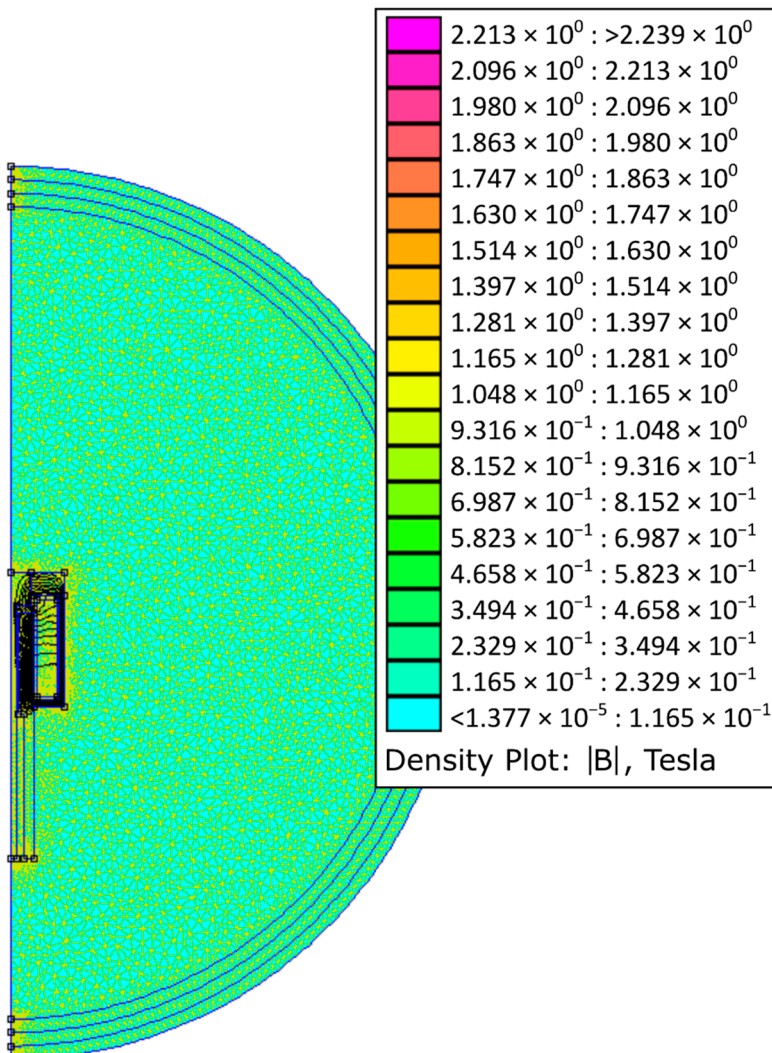

**Figure 12.** Simulation of the attractive force of an electromagnet: the boundary condition.

Figure 13 on the left shows the model in the FEMM 4.20 program with the finite element network and defined materials; the right part of the model, after simulation in the postprocessor, shows the magnetic flux lines and colour rendering of the magnetic field intensity, where purple represents the highest intensity. The carcass material, which is in fact plastic, is replaced by air because they have similar magnetic properties.

Figure 14 shows the simulation results for the initial state when the air gap is zero. Figure 15 shows the simulation results for the maximum air gap condition.

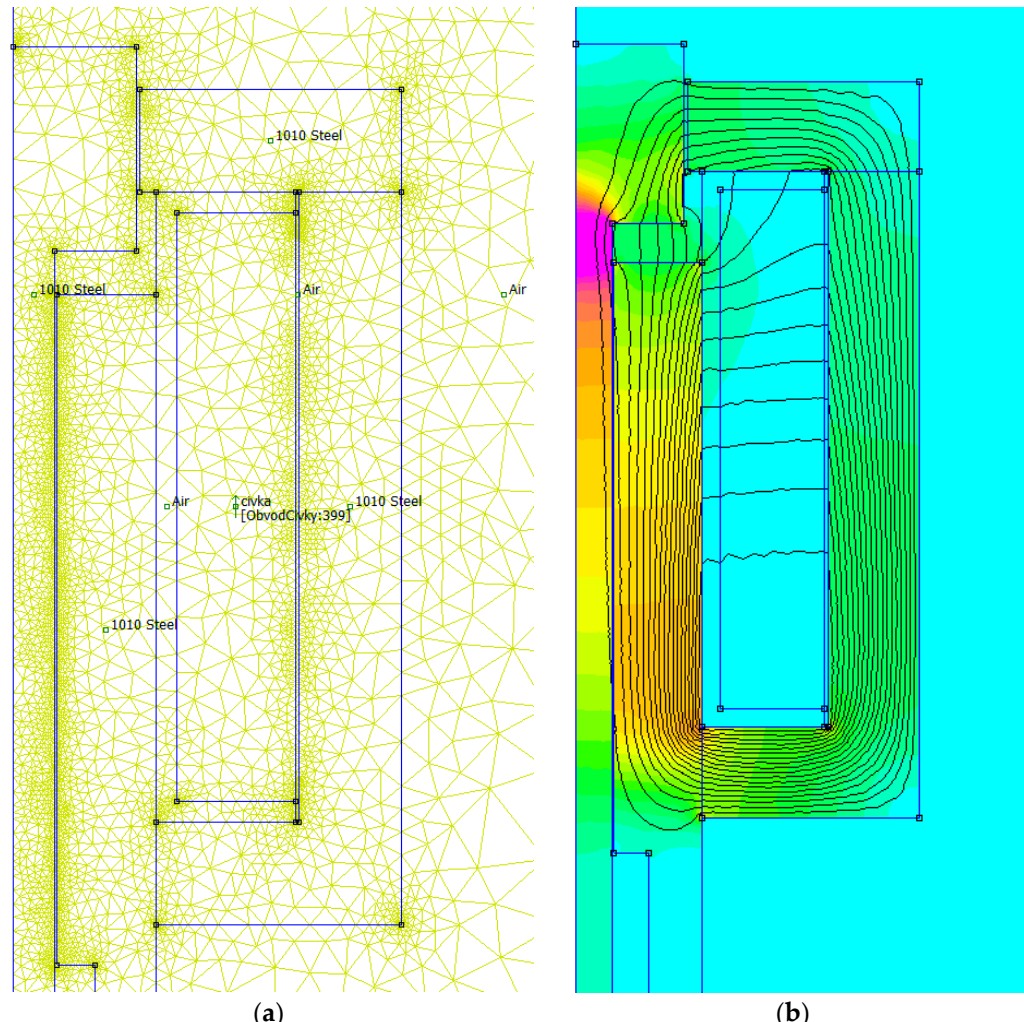

**Figure 13.** Simulation of the attractive force of an electromagnet: (**a**) mathematical model with created simulation network; (**b**) simulation result with shown magnetic flux lines and magnetic field intensity.

As already mentioned, the main output of the simulation is the characteristic of the attractive force. Figure 16 shows the characteristics for the two materials mentioned above. It can be seen that the AISI 1010 steel has a more linear course than the Supermaloy steel, which has a sharp break in its air gap area of 0.5 mm but a 50% greater attractive force in the attracted state. The core material AISI 1010 or DIN 1.0413 was chosen for its design, due to greater material availability, lower price and also because the electromagnet from this material has a lower inductance of 0.033 H than the electromagnet from Supermaloy, of 0.068 H.

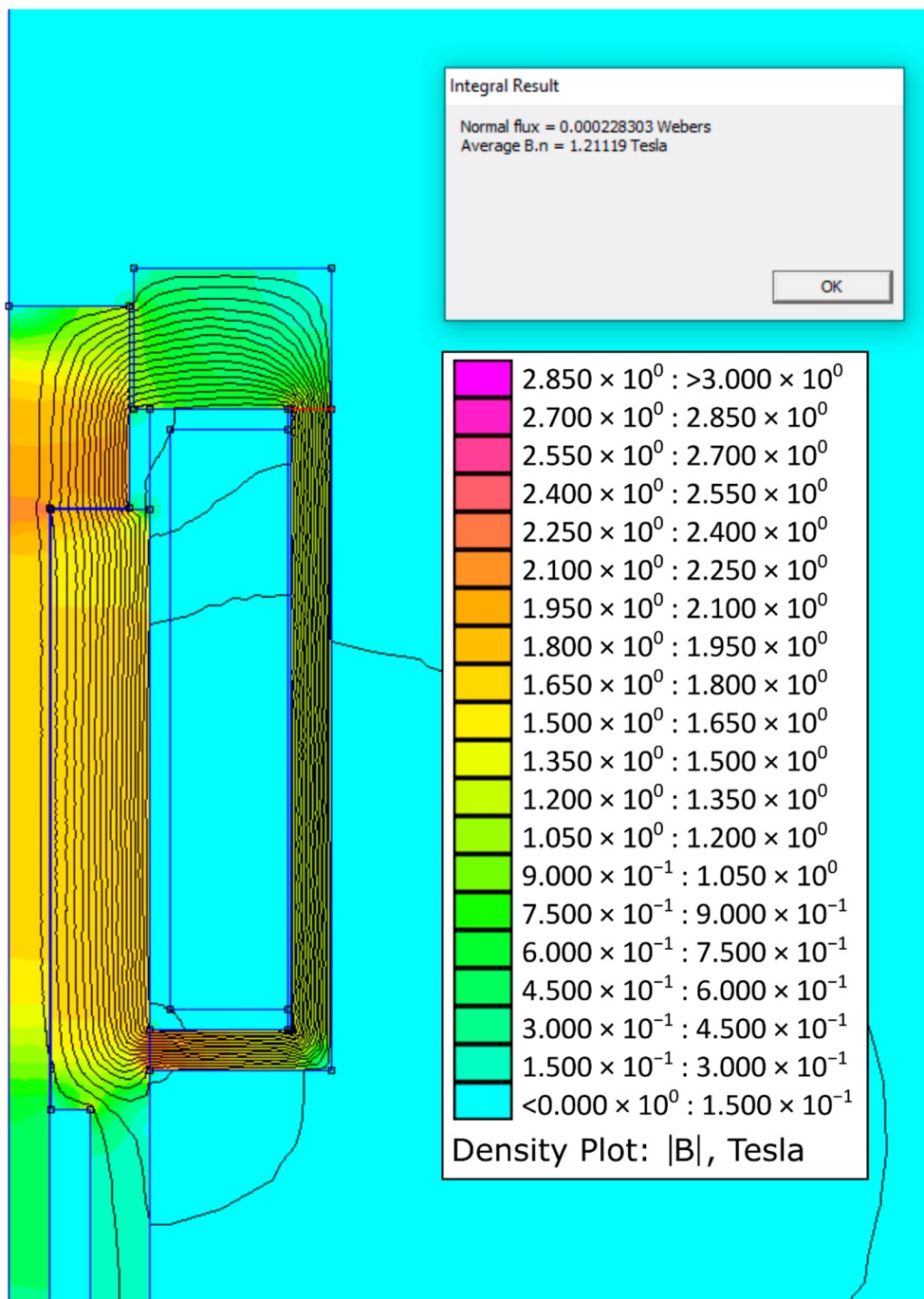

**Figure 14.** Distribution of magnetic flux and magnetic field intensity for minimum air gap.

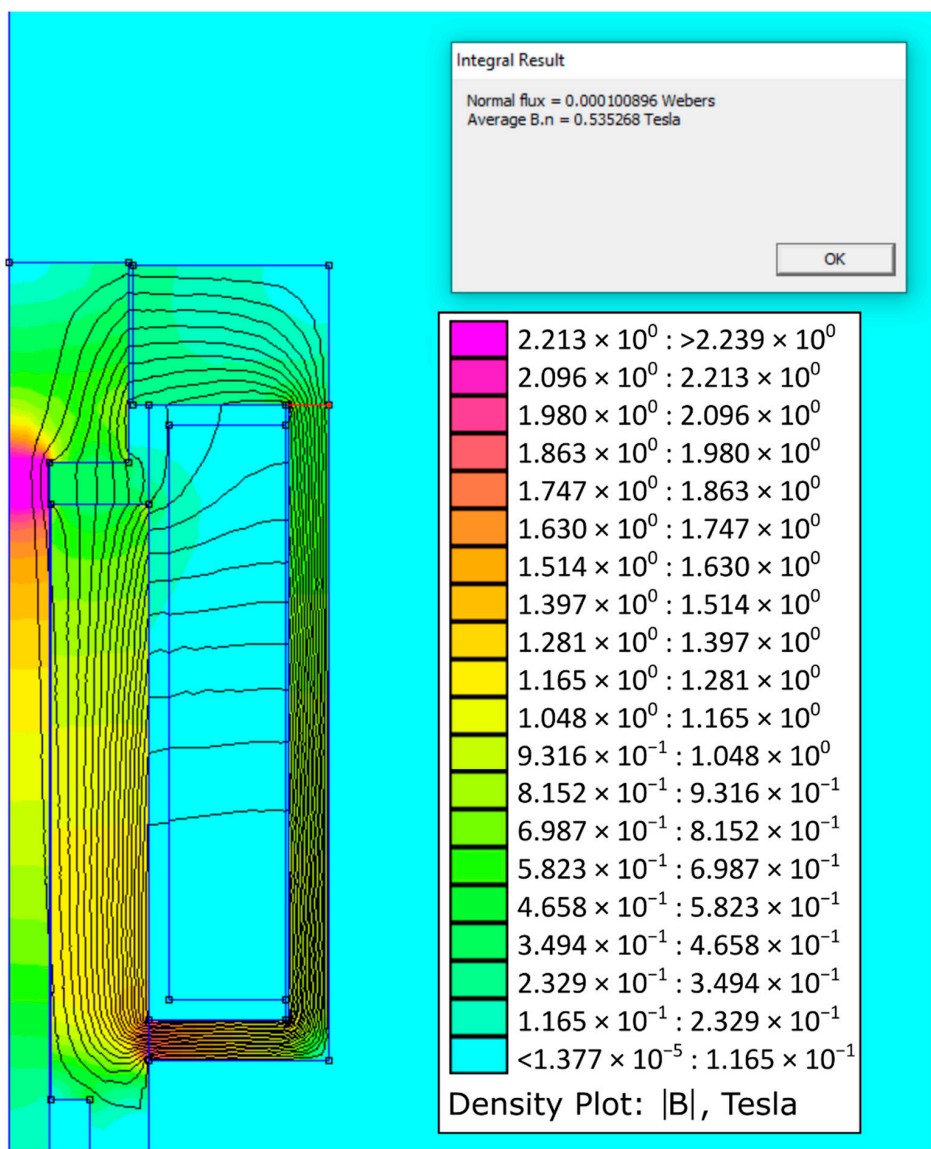

**Figure 15.** Distribution of magnetic flux and magnetic field intensity for maximum air gap.

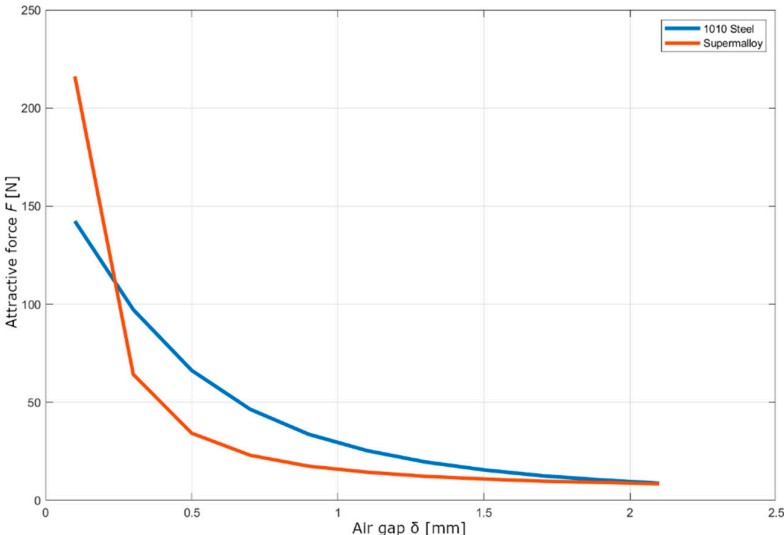

**Figure 16.** Courses of the electromagnet attractive force obtained from the simulation.

### 3.4. Coil Electrical Circuit Design

Because the electromagnet operates with relatively high dynamics, the transients of the coil are important, especially the dependence of the current flowing through the magnetic field on time and voltage. In general, the higher the voltage, the faster the saturation of the magnetic circuit, but if the time of increased voltage is too long, the coil heats up excessively. Therefore, it is necessary to determine how long it takes for the circuit to saturate.

From the previous simulation we know the value of the inductance of the electromagnet, i.e., the inductance of the coil with the inserted soft core, so we can use the following equation to describe the current and voltage:

$$U = L\frac{dI}{dt} + RI \tag{11}$$

Assuming that the resistivity and inductance are constant, the equation can be adjusted to these forms by time derivation:

$$\dot{I} + \frac{RI}{L} - \frac{U}{L} = 0 \tag{12}$$

$$\dot{I} = \frac{U - RI}{L} \tag{13}$$

The differential equation [27] can then be solved using a homogeneous solution:

$$I(t) = C_1 + C_2 e^{-(\frac{R}{L})t} \tag{14}$$

There is defining initial and final coil saturation conditions. At the beginning, there is zero current; at infinity, the current approaches the constant value given by Ohm's law:

$$I = \frac{U}{R} \tag{15}$$

$$I(0) = 0 = C_1 + C_2 \times 1 \tag{16}$$

$$C_1 = -C_2 \tag{17}$$

$$\lim_{(t \to \infty)} I = \frac{U_{\text{stat}}}{R} = -C_2 + C_2 e^{-(\frac{R}{L})T} = C_2\left(-1 + e^{-(\frac{R}{L})\infty}\right) \tag{18}$$

$$C_2 = \frac{U_\infty}{R} \tag{19}$$

From which, the constants $C_1$ and $C_2$ can be calculated and substituted into the solution:

$$I(t) = \frac{U_{\text{stat}}}{R}\left(1 - e^{-(\frac{R}{L})t}\right) \tag{20}$$

In this case, we are interested in how long the current value approaches the asymptote $\frac{U_{\text{stat}}}{R}$. The value of the approximation can be defined, for example, as:

$$C_{95} = 0.95 \tag{21}$$

$$I_{95} = C_{95}I_\infty = C_{95}\frac{U_{\text{stat}}}{R} \tag{22}$$

so we are looking for a current that has a value of 95% of the constant current value.

The equation is, therefore:

$$C_{95}\frac{U_{\text{stat}}}{R} = \frac{U_{\text{stat}}}{R}\left(1 - e^{-(\frac{R}{L})T}\right) \tag{23}$$

where $T$ is the time sought when the coil will be sufficiently saturated:

$$T = -\frac{L}{R}\ln(1 - C_{95}) \tag{24}$$

These equations apply to the case when the voltage value is equal to the static voltage value at which the coil is rated. However, for faster saturation, it is possible to increase the voltage value after a while. After that, however, the equations compiled above no longer apply, as this is a discontinuous problem. We can partially approach the desired value of the saturation time if we adjust the current equation to the following form:

$$\frac{U_{\text{stat}}}{R} = \frac{U}{R}\left(1 - e^{-\left(\frac{R}{L}\right)T}\right) \tag{25}$$

$$T = -\frac{L}{R}\ln\left(\frac{U - U_{\text{stat}}}{U}\right) \tag{26}$$

when the value of the asymptote approximation remains the same, but the course of the exponential curve is adjusted to the new voltage value. The asymptote approximation value is no longer needed because the new curve intersects the static current value.

Instead of an analytical calculation, it is also possible to calculate the original differential equation in the Matlab program, with the current set point.

To describe the state of discharge of the coil, we can use a slightly modified equation, where we will again be interested in the time in which a certain current value is reached; in this case, when it will be equal to zero. Here, too, the principle can be used to use a higher voltage than the nominal one for the transient to speed up the process. However, due to the polarity of the previous charge, it is now necessary to connect a negative voltage that is a multiple of the nominal static voltage of the coil:

$$I(t) = \frac{U_{\text{stat}}}{R}\left(e^{-\left(\frac{R}{L}\right)t}\right) \tag{27}$$

All of the above waveforms can be seen in the graph in Figure 17. The current functions during charging and discharging the coil have the same shape, only mirrored and offset along the horizontal axis. It can be seen that at a higher charging voltage $2U_{ref}$, a faster charge will occur in time $t_{n2}$ than at a nominal voltage $t_{n1}$. The situation is the same for discharging, only due to the initial conditions, it is necessary to move the discharge at higher voltages along the horizontal axis so that at time $t = 0$, the current value is equal to the static value of the coil current. The graph also plots the values of sufficient approximation to the current asymptote, one for charging and the other for discharging the coil.

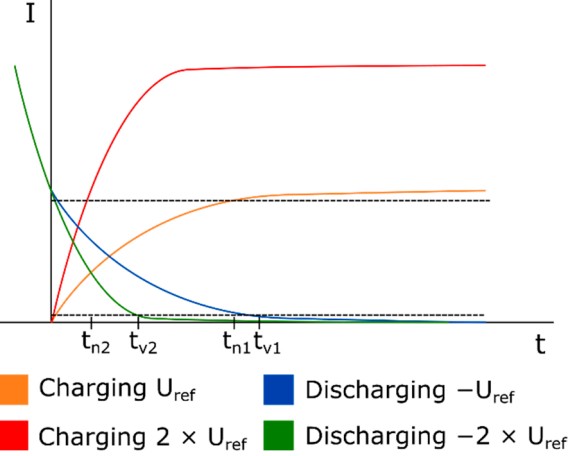

**Figure 17.** Courses of current during charging and discharging at different voltages.

The resulting selected voltage profile can be similar to Figure 18; this is a solution where the electromagnet is in a pushing arrangement. The voltage changes abruptly depending on the position of the cam; in the mechanism gripped state, the current is reduced, as not all the force that would be generated by the electromagnet is needed. The reduction in the voltage will bring a smaller value of the generated waste heat in the conductor. To release the needle, it is necessary to demagnetize the entire magnetic circuit, so a negative voltage is applied during demagnetization. Thereafter, the coil is de-energized until just before the point of gripping the needle, when a voltage multiplier is applied, which results in a sharp increase in current. At this point, it is possible to briefly exceed the rated current of the coil to create a value stronger than the rated force of the electromagnet. Then, the stress is reduced again to the initial value; between these two states, it is possible to insert another intermediate stage to achieve the desired characteristics of the attractive force.

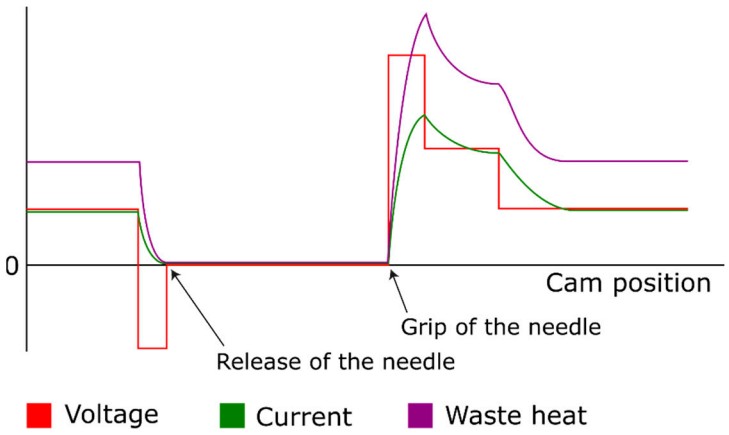

**Figure 18.** Courses of physical quantities of the coil.

The ideal source with a hard characteristic, i.e., with a constant voltage at different loads, is chosen as the voltage source. The voltage of the source is externally controlled so that the optimal behaviour of the whole needle bar system is achieved. It can be assumed that in the real case, the source will not have the ideal designed characteristics and will contain delays and instabilities, which will bring a certain amount of deviation into the system, which must be taken into account.

## 4. Discussion

The current design of the mechanical system of the needle bar of the special sewing machine imitating a hand stitch makes it impossible to increase the operating speed of the sewing machine due to the increase in vibration and noise. Scientific work has been done in the past focused on needle bar innovation to reduce these problems. This work deals with the design of a completely new principle of needle gripping and releasing control using an electromagnet located in the body of the needle bar.

In the first part, the type of electromagnet was chosen together with the design of its parameters and dimensions according to the typically used design procedure. The result of the preliminary design is a push solenoid, which has a working stroke of 2 mm and an attractive force at a maximum air gap of 6 N, supplied by a controlled direct current of 3 V, but with the possibility of overload to 24 V as the armature moves.

In the next part, the computational model of the electromagnet was built in the finite element program FEMM 4.20 and the characteristics of the attractive force, depending on the size of the air gap or the position of the armature in the working range, were determined. Since the empirical design of the solenoid parameters is valid for the pull solenoid, the calculated forces in the FEMM 4.20 program differed due to the different design; specifically for push solenoids, the force is greater at maximum air gap, which was confirmed by calculation of 8.2 N.

It is clear from the design of the electromagnet that this could be the right way to create an alternative means to control the needle grip and release in the needle bar. However, for the correct design of the electromagnet and the theoretical verification of its functionality, it will be necessary to create several more numerical computational models that describe the various aspects of the needle bar operation. The plan is to create, in parallel with the above-mentioned models, a complex model for the Matlab Simulink environment. This model should describe the mechanical parts of the needle bar and the bonds between them, the kinematic excitation of the needle bar shell, the electrical circuit that supplies the solenoid, and the solenoid itself, which is located between the shell and the core. It will be possible to apply a number of optimization tasks to this complex model. For example, the design of return spring parameters that will exert sufficient force to meet the required dynamics but will not unnecessarily load the mechanism with excessive force in the expanded state. It will also be necessary to design a suitable power supply for the electromagnet, as the gripping of the needle must be realized in the order of milliseconds. It will be necessary to design the coil overvoltage from nominal 3 V to 24 V in order to saturate the magnetic circuit faster. When demagnetizing, a negative voltage of $-24$ V will be used for a faster collapse of the magnetic field and, thus, the release of the needle. It will therefore be necessary to determine the durations of this overvoltage or the application of a negative voltage in order to avoid overloading the coil or creating the opposite magnetic field.

The proposed system could most likely lead to a reduction in vibrations and noise. The basic assumption is that the vibrations caused by the original control of the needle release, which uses the impact of the control element on the stopper, will be completely eliminated, because this very significant source of vibrations will be replaced by a newly designed magnetic system. That is, this current source of vibration will be completely eliminated. Nevertheless, of course, we do not expect the vibrations of the magnetic control of needle attachment to be zero, but they will be significantly lower. Evidence will be provided when the research is in the phase when a functional model of the developed magnetic system is produced; then, the measurement of actual noise values will be performed and compared with the original system.

This innovation could also allow the machine to increase operating speed and, thus, productivity. Due to the use of a higher charging voltage, the time required to attach and release the needle can be reduced. However, it is also possible to design an electromagnetic circuit with respect to higher speeds of the needle bar.

There is also a noticeable support for automation in the field of the clothing industry. The new needle bar, together with the previously designed needle bar servo drive system, could facilitate the transition of the sewing machine to automated production.

Moreover, the structure is also interesting for intelligent sewing machines that are under development. Intelligent sewing machines have the ability to automatically adapt to changes in the technological processes of sewing clothes. Therefore, the presented needle attachment structure could be one of the solutions when needles are changed on other sewing machines due to the sewing of very different materials from which clothes are made; these are frequent cases in modern clothing production processes.

In this way, this research activity responds to the pressure of the consumer industry to reduce production costs and, at the same time, seeks to support the automation of the sewing process and also to ensure greater safety for machine operators by reducing noise levels.

**Author Contributions:** Conceptualization, J.K. and V.K.; methodology, J.K. and V.K.; software, V.K.; validation, V.K.; formal analysis, V.K.; investigation, J.K. and V.K.; resources, J.K. and V.K.; data curation, J.K. and V.K.; writing—original draft preparation, V.K.; writing—review and editing, J.K.; visualization, J.K.; supervision, J.K.; project administration, J.K.; funding acquisition, J.K. All authors have read and agreed to the published version of the manuscript.

**Funding:** This research was funded by the Student Grant Competition of the Technical University of Liberec under the project No. SGS-2022-5046.

**Conflicts of Interest:** The authors declare no conflict of interest. The funders had no role in the design of the study, in the collection, analyses, or interpretation of data, in the writing of the manuscript, or in the decision to publish the results.

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
