# Peer review of "Design of Electromagnetic Control of the Needle Gripping Mechanism"

_machines, doi:10.3390/machines10050309_

Round 1

Reviewer 1 Report

The article deals with the current topic of electromagnetic control of the needle gripper mechanism of sewing machines.

The authors describe in detail conventional machine mechanisms on the hearts for sewing a decorative stitch or imitating hand sewing stitches.

The authors propose to replace the machine mechanism with an electromagnetic mechanism that is smaller, faster, and produces less noise.

The article describes in detail the proposed mechanical structure, the electromagnetic coils and the simulation of the attraction force and other structural parameters.

The structure is very well designed and is very instructive for other researchers working in this field.

The structure is very interesting for intelligent sewing machines that are under development. Intelligent sewing machines have the ability to automatically adapt to changes in technological processes of sewing clothes, so the presented structure could be one of the solutions when needles are changed on other sewing machines due to sewing of very different materials from which clothes are made, and these are frequent cases in modern clothing production processes.

I think it would be good if the authors listed this as an option in the introductory part of the article, which would further increase the importance and value of the article.

I recommend the publication of the article.

Author Response

Dear reviewer,

Thank you for your review. We agree with your comments. At our department, we deal with the issue of automation of the sewing process and we really believe that it would be possible to use this method for automatic needle exchange.

We will add your recommendation to the manuscript.

Kind regards,

Authors

Reviewer 2 Report

Modeling of a sewing machine is presented.  The analytical model is sound.  The results presented demonstrated the efficiency of the developed model.  Some suggestions can be taken in to consideration.

  1. A force displacement curve to illustrate the pulling/pushing force should be inserted in figures 3-6.
  2. “the contact force must be negative for the closed state and positive for the open state”. Contact force can be zero with no contact.
  3. What are the Refs for equations 2-9?
  4. How are the values in Table 2 obtained?
  5. Boundary conditions for the finite element analyses should be plotted in a figure.
  6. Mesh convergence study may be presented.
  7. Coupling between mechanical simulation and magnetic simulation may be required, which is not considered in the paper.
  8. Verification of the developed model may be needed. If possible, compare the analytical results with the finite element analyses.

Author Response

Dear reviewer,

Thank you for your comments. We really appreciate the time you dedicated to us and we would like to respond to your comments in this way.

Point 1: A force displacement curve to illustrate the pulling/pushing force should be inserted in figures 3-6.

Response 1: We agree that it would be interesting to state these characteristics, but unfortunately a detailed analysis of these forces has not been performed. When choosing the type of electromagnet, we based on generally known assumptions, when it is most advantageous to use the solenoid type configuration to realize the motion of the core. For other arrangements, the required force would only be achieved in the attracted state.

Point 2: “the contact force must be negative for the closed state and positive for the open state”. Contact force can be zero with no contact.

Response 2: We agree with your comment. We will add your recommendation to the manuscript.

Point 3: What are the Refs for equations 2-9?

Response 3: These values were given together with the calculated parameters in Table 1, but this was not very understandable, so we preferred to divide the table. The reference parameters are now separated in Table 1, see modified manuscript.

Point 4: How are the values in Table 2 obtained?

Response 4: The table contains the construction dimensions of the electromagnet, which is based on the technological possibilities of electromagnet production and the construction arrangement of the needle bar. The basic parameters obtained on the basis of the calculation given in chapter 3.2 were respected during the design.

Point 5: Boundary conditions for the finite element analyses should be plotted in a figure.

Response 5: Thank you for your comment. The boundary condition is set to "free environment", interference with the needle bar is not considered. We will add the information, including the picture, to the manuscript.

Point 6: Mesh convergence study may be presented.

Response 6: Unfortunately, the FEMM software does not allow this.

Point 7: Coupling between mechanical simulation and magnetic simulation may be required, which is not considered in the paper.

Response 7: Yes, we are aware of that. The mechanical simulation is in the plan of the proposed following procedure. As we mentioned in the discussion.

Point 8: Verification of the developed model may be needed. If possible, compare the analytical results with the finite element analyses.

Response 8: We agree that verification of the developed model may be necessary. In this work, an electromagnet was designed for a nominal force of 6 N at an air gap of 2 mm. This can be theoretically confirmed retrospectively using formula (1). However, this is relatively inaccurate. The result of the simulation was a force of 8.2 N at an air gap of 2 mm, as mentioned in the discussion.

If interested, we are open to further discussion.

Thank you.

Kind regards,

Authors

Reviewer 3 Report

REVIEW REPORT

TITLE: Design of electromagnetic control of the needle gripping mechanism

Manuscript ID:      machines-1684871

Section:                  Mechatronic and Intelligent Machines

Special Issue:         Advances in Computer-Aided Technology

  1. The needle must be held with a minimum force 92 of 25 N when piercing. The force must be at least 15 N during drawing through the sewn 93 material. On what basis these values are decided any reference or calculation must be there.

  1. most basic electromagnetic arrangement is the front electromagnet (Figure 3). LINE 108. Remove the bracket as it is not there for other figures. And also (Figure 4) has a flatter force characteristic, LINE 118. Other places also throughout the manuscript correct it.

  1. In Figure 11, the dimensions of the electromagnet has been given elaborately, but the optimum dimension’s required for electromagnet effectiveness by considering the Air gap, Current and Flux etc., must be included to highlight the design of the needle gripper.

  1. Regression Analysis must be given to by considering the design factors and the output parameters in the current selected parameters under Magnetized and De-Magnetized state.

  1. Please give the FEMM 4.20 program Simulation results for the Maximum and Minimum Flux induced due to air gap in the optimum design of an electromagnet.

  1. In the LINE 398, the calculated forces in the FEMM 4.20 program differed due to the different design, specifically for push solenoids the force is greater at maximum air gap, which was confirmed 399 by calculation of 8.2 N. So, at this point what is the amount of Vibration observed and Noise observed.

  1. More over in LINE 402 onwards the authors are presenting that could most likely lead to a reduction of vibrations, as the release and grip of the needle are controlled electromagnetically. This innovative way could also allow the machine to increase operating speed and thus productivity. Please elaborate in amount of quantification of Reduction of vibration and Noise in possible cases of the efficient design and its relation to the gripper force and other design parameters.

Overall effort by AUTHORS are appreciated and also Good
Representation of Design of electromagnetic control of the needle gripping mechanism Authors elaborately discussed design factors related to the development and implementation of needle gripping mechanism. The Paper can be accepted after incorporating the comments in the Manuscript.

Author Response

Dear reviewer,

Thank you for your comments. We really appreciate the time you dedicated to us and we would like to respond to your comments in this way.

Point 1: The needle must be held with a minimum force 92 of 25 N when piercing. The force must be at least 15 N during drawing through the sewn 93 material. On what basis these values are decided any reference or calculation must be there.

Response 1: These values were determined by measuring the forces acting on the needle during the sewing process. We will add the reference to this work to the manuscript.

Point 2: most basic electromagnetic arrangement is the front electromagnet (Figure 3). LINE 108. Remove the bracket as it is not there for other figures. And also (Figure 4) has a flatter force characteristic, LINE 118. Other places also throughout the manuscript correct it.

Response 2: We agree with your comments. We will add your recommendation to the manuscript.

Point 3: In Figure 11, the dimensions of the electromagnet has been given elaborately, but the optimum dimension’s required for electromagnet effectiveness by considering the Air gap, Current and Flux etc., must be included to highlight the design of the needle gripper.

Response 3: Sorry, we do not fully understand this recommendation.

Point 4: Regression Analysis must be given to by considering the design factors and the output parameters in the current selected parameters under Magnetized and De-Magnetized state.

Response 4: We agree with your opinion. The dimensions of the electromagnet have now been determined with respect to the design of the needle bar. The main parameters that can be affected are the length ratio to armature diameter and the ratio of winding thickness to armature diameter, which were chosen based on the ratio of core weight to coil weight. More detailed optimization is planned in the future work, where the coil will be optimized with respect to the required times of magnetization and demagnetization.

Point 5: Please give the FEMM 4.20 program Simulation results for the Maximum and Minimum Flux induced due to air gap in the optimum design of an electromagnet.

Response 5: We will add the mentioned results to the manuscript.

Point 6: In the LINE 398, the calculated forces in the FEMM 4.20 program differed due to the different design, specifically for push solenoids the force is greater at maximum air gap, which was confirmed 399 by calculation of 8.2 N. So, at this point what is the amount of Vibration observed and Noise observed.

Point 7: More over in LINE 402 onwards the authors are presenting that could most likely lead to a reduction of vibrations, as the release and grip of the needle are controlled electromagnetically. This innovative way could also allow the machine to increase operating speed and thus productivity. Please elaborate in amount of quantification of Reduction of vibration and Noise in possible cases of the efficient design and its relation to the gripper force and other design parameters.

Response 6 and 7: Vibration values ​​were not monitored in this part of the work. The basic assumption is that the vibrations caused by the mechanical control of the needle release will be completely eliminated, because this very significant source of vibrations will be replaced by a magnetic system. That is, this source of vibration will be completely eliminated. Nevertheless, of course, we do not expect the vibrations of the magnetic needle attachment to be zero, but they will be significantly lower. It is really planned to present significant values ​​for reducing vibration and noise, but only when a functional model of a new needle bar with magnetic control is built. We are prepared for this, because noise measurements were previously performed on a functional model of the needle bar movement mechanism, where the values ​​of sound intensity and sound pressure of the original needle bar and new needle bar with the proposed modifications were compared. When the research is in the phase when a functional model of the developed magnetic system is produced, we really aim to measure its actual noise values ​​and compare them with the original system.

If interested, we are open to further discussion.

Thank you.

Kind regards,

Authors